# A Review on Micro-LED Display Integrating Metasurface Structures

**DOI:** 10.3390/mi14071354

**Published:** 2023-06-30

**Authors:** Zhaoyong Liu, Kailin Ren, Gaoyu Dai, Jianhua Zhang

**Affiliations:** 1School of Microelectronics, Shanghai University, Shanghai 200444, China; lzyly@shu.edu.cn (Z.L.); renkailin@shu.edu.cn (K.R.); gaoyudai@shu.edu.cn (G.D.); 2Key Laboratory of Advanced Display and System Applications (Ministry of Education), Shanghai University, Shanghai 200444, China; 3Shanghai Key Laboratory of Chips and Systems for Intelligent Connected Vehicle, Shanghai University, Shanghai 200444, China

**Keywords:** Micro-LED, metasurface, light extraction efficiency, angular deflection, polarization

## Abstract

Micro-LED display technology has been considered a promising candidate for near-eye display applications owing to its superior performance, such as having high brightness, high resolution, and high contrast. However, the realization of polarized and high-efficiency light extraction from Micro-LED arrays is still a significant problem to be addressed. Recently, by exploiting the capability of metasurfaces in wavefront modulation, researchers have achieved many excellent results by integrating metasurface structures with Micro-LEDs, including improving the light extraction efficiency, controlling the emission angle to achieve directional emission, and obtaining polarized Micro-LEDs. In this paper, recent progressions on Micro-LEDs integrated with metasurface structures are reviewed in the above three aspects, and the similar applications of metasurface structures in organic LEDs, quantum dot LEDs, and perovskite LEDs are also summarized.

## 1. Introduction

Micro-LEDs have attracted much attention owing to their advantages of high luminous intensity, high resolution, high contrast, fast response speed, long lifespan, and low power consumption. Due to these excellent performance traits, Micro-LEDs are regarded as the mainstream of next-generation display technology for a wide range of applications, from wearable devices such as wristbands and watches to commercial billboards, public displays, and virtual reality (VR) or augmented reality (AR) devices [1,2,3,4,5]. However, challenges have also arisen with the development of Micro-LED display technology, such as mass transfer, full-color display, and size-dependent efficiency [6,7]. The luminous efficiency of Micro-LEDs decreases rapidly as the size decreases, so it is necessary to improve the light extraction efficiency (LEE) to improve the external quantum efficiency (EQE) [8]. Nowadays, there are many methods to improve LEE. This paper mainly reviews the approaches to integrating metasurface structures on Micro-LEDs.

The metasurface is an artificial nanostructure that is designed to control the amplitude, polarization, and phase of incident waves at the subwavelength scale [9,10,11,12]. Metasurface structures can realize the above functions with the premise that the incident light must be coherent [13]. However, a typical Micro-LED exhibits Lambertian-shaped emission [14]. Light emitted in any direction has very low spatial coherence, so it is a key issue to realize control of the Micro-LED wavefront with metasurface. Therefore, the researchers introduced reflective mirrors at the bottom and top of the Micro-LED to form a Fabry–Perot (F-P) cavity structure [13], so that the emitted light is concentrated in a narrow angular range after resonance selection through the cavity, which can enhance the spatial coherence of the emitted light, and the collimation of the emitted light also improves the LEE. In this case, the integration with the metasurface structures can realize the deflection of the beam angle, and the light can be emitted to the preset position to fully utilize the emitted light.

In addition, Micro-LEDs that emit polarized light play a key role in near-eye displays, but obtaining polarized emission from LEDs requires complex design and manufacturing [15]. Moreover, obtaining polarized light emission is difficult due to the weak anisotropy of Micro-LEDs, so wave plates are needed to improve the anisotropy of Micro-LEDs. However, traditional wave plates are not conducive to Micro-LED integration due to their large size. The appearance of the metasurface structures solves these problems because of their small size and simple implementation process [16]. Moreover, the combination of metasurface structures and optical gratings can realize linear and circular polarization. This paper reviews the research progress of Micro-LEDs integrated with metasurface structures in improving the LEE, collimation, controlling angle deflection, and controlling polarization. The latest research results in the four directions mentioned above are demonstrated in Figure 1.

## 2. Improvement in Light Extraction Efficiency

As one of the important performance indicators of Micro-LEDs, EQE refers to the ratio of the final emitted photon number to the injected carrier number, which can be obtained by the product of the LEE and internal quantum efficiency (IQE). Therefore, EQE can be improved by improving the LEE. Currently, methods to improve the LEE of Micro-LEDs include flip chip technology, transparent substrate technology, patterned substrate technology, surface microstructure technology, and bottom reflector technology [20,21,22]. This paper mainly introduces metasurface structures to improve the LEE and summarizes them in this chapter (listed in Table 1).

### 2.1. Improvement in the LEE by Metasurface Structures

Since the refractive index of the surface material of Micro-LEDs is much larger than that of air, the full reflection angle of the light emitted by Micro-LEDs into the air is very small, causing total reflection phenomena to very easily appear, which is also one of the reasons for the low LEE [30]. The integrated metasurface structures on the top surface of Micro-LEDs can not only effectively expand the light emitting area of Micro-LEDs, but also change the incident angle of light on the inner surface of the semiconductor, thus destroying the total reflection condition of part of the light so as to significantly improve the LEE.

Mao et al. proposed disordered metasurface LEDs by studying the distribution and size of the cuticle fringes in fireflies [17]. The metasurface structure changed from an orderly arrangement of square gratings to a curved top surface, then to a disordered arrangement, and was finally designed as Ag nanoparticles with a curved top surface and disordered arrangement (see Figure 2a). LEDs with Ag-free nanoparticles were compared with those with Ag nanoparticles. Figure 2b,c show the photoluminescence (PL) and electroluminescence (EL) spectra with and without Ag nanoparticles, respectively. The results showed that Ag nanoparticles enhanced PL and EL by 170% and 140%, respectively. The wavelength of the experiment in this paper can also be obtained from Figure 2b,c by taking the full width at half-maximum (FWHM); the wavelength is found to be 440~470 nm, and it is listed in Table 1. The wavelengths of all tables in this review are derived from this. When the working wavelength is 452 nm, the output power of the LED with Ag nanoparticles is increased by 2.7 times, and the absolute EQE is increased from 31% to 51% (see Figure 2d,e). In addition, the top metasurface structure can be made in one step by the gas cluster technology, which reduces the complexity of the process and has a good application prospect.

In perovskite LEDs (PeLEDs), nanobricks were embedded in electron transport layers to enhance the LEE [23]. This paper presented that the nanopatterned PeLEDs not only improved the LEE, but also showed significant directional emission. The fundamental reason is that due to the diffraction effect of the nanopattern of the medium, more photons fall into the escape cone, resulting in a directional emission pattern. The far-field intensity of PeLEDs with Ag nanopatterns was significantly increased by eight times compared with planar PeLEDs. The optical power loss ratios of planar and nanopatterned PeLEDs at 780 nm wavelength were investigated. The ratios presented that the waveguide mode caused by the large refractive index difference between the active layer and ITO layer is an important factor limiting the LEE.

### 2.2. Improvement in the LEE by Metasurface Structure in Organic LEDs (OLEDs) or Micro-OLEDs

Metasurfaces are also widely used in OLEDs or Micro-OLEDs to improve the LEE. In 2016, Zhou et al. proposed integrating a speckle image holography (SIH) metasurface at the top to obtain OLEDs with high contrast and high efficiency [24], as shown in Figure 3a. The SIH metasurface, due to its “regional characteristics”, shown in Figure 3b, has a wide viewing angle and high contrast directional gain that allows control of the non-interference wavefront generated by the emission layer. Compared with OLEDs with one-dimensional gratings integrated at the top, the efficiency improvement in the SIH metasurface is not affected by wavelength, while the enhanced LEE of the one-dimensional gratings is dependent on wavelength and angle due to resonance (see Figure 3c). The power and EQE improvement in the SIH metasurface are 1.5 times and 1.4 times of one-dimensional gratings, respectively. Next year, by experimenting with three different OLEDs, fluorescent green OLEDs, phosphorescent red OLEDs, and phosphorescent blue OLEDs, the team compared the results of SIH metasurface OLEDs with one-dimensional gratings and flat OLEDs, and found the same experimental results. These results demonstrated that the SIH metasurface improved the LEE and light control independent of material, wavelength, and radiation angle [29].

In OLEDs, in addition to the SIH metasurface, supergrating structure is also a common method to improve the LEE of OLEDs [25,26]. Compared with OLEDs without supergratings, the light output intensity of supergrating OLEDs is 4.8 times higher at 510 nm. The OLED structure with supergratings is shown in Figure 4a. The reflected supergratings effectively couple the beam captured in the waveguide mode to enhance the LEE. The team compared the effects of periodic and quasi-periodic supergratings on the luminescence intensity of OLEDs and found that although periodic supergratings are enhanced more, the polarization dependence is very strong. Therefore, periodic supergratings need to be replaced by quasi-periodic supergratings in some specific applications. Kang et al. reported a nanoslot metasurface enhances the LEE of OLEDs [27]. As shown in Figure 4b, the structural parameters of nanoslots include width W and length L1 and L2. The layer cross-section of nanoslot metasurface OLEDs and flat OLEDs are shown in Figure 4c. The addition of nanoslot metasurface can induce surface plasmon (SP) and localized SP mechanisms to enhance the external coupling efficiency and reduce the ambient light reflectance of OLEDs. The change in metasurface layer thickness has an influence on both external coupling efficiency and environmental reflectance; thus, the properties of both should be considered to strike a balance when changing the thickness. The structural parameters of nanoslots have little effect on the external coupling efficiency, but have a great effect on the reflectance. Therefore, a small reflectance can be obtained by adjusting the structural parameters of nanoslots. The performance of the optimized nanoslot metasurface OLEDs is 15% higher than that of the traditional flat OLEDs.

In Micro-OLEDs, Lin et al. proposed using metalens to enhance the LEE, as shown in Figure 5a [28]. Through simulations, it was found that metalens can convert different wave vectors into directly emitted wave vectors in a single interaction with light, thereby improving the LEE of Micro-OLEDs. It is worth noting that metalens does not require multiple interactions with photons, i.e., multiple reflections are not needed. Therefore, metalens exhibits strong optical coupling effects. As shown in Figure 5b,c, when the focal length is between 1.6 and 2.2, Micro-OLEDs with metalens exhibit a significant improvement in EQE compared to traditional Micro-OLEDs. These technologies are expected to boost AR/VR.

## 3. Improvement in the Emitted Light Collimation

In micro-displays composed of pixel arrays formed by Micro-LEDs, the close proximity of pixels and the Lambertian-shaped emission of Micro-LEDs result in optical crosstalk between adjacent pixels, affecting display clarity. Improving the collimation of Micro-LED light emission is beneficial for reducing optical crosstalk between adjacent pixels and improving display performance. Furthermore, metasurface structures can only modulate light with strong spatial coherence, so it is necessary to improve the collimation of Micro-LEDs in order to be controlled by metasurfaces and generate more functionalities. Currently, methods to improve the collimation of Micro-LEDs include resonant cavities (RC) [31], plasma collimation [32,33], lens collimation [34,35], and light-blocking patterns between pixels [36]. However, methods like using light-blocking patterns between pixels achieve overall collimation of Micro-LEDs rather than collimation of the Micro-LEDs’ intrinsic emission. Therefore, such methods cannot achieve metasurface control over Micro-LEDs. Currently, the most mainstream method to achieve metasurface control over the phase of Micro-LED emission is through RC. The recent research progress in improving the collimation of LEDs is summarized in Table 2.

Additionally, the thickness of OLEDs and PeLEDs is only a few hundred nanometers, while Micro-LEDs are a few micrometers. Therefore, the interference effect of the F-P cavity is more obvious in OLEDs and PeLEDs. Moreover, due to the difference in device structure, the emission position of Micro-LEDs is within the multi-quantum wells (MQWs), resulting in different heights of the light sources, which makes the microcavity effect more complex. For some Mcro-LEDs with strong cavity effects, this is mainly because they apply thin-film Micro-LEDs. It should be noted that the design of the top and bottom mirrors of the RC is related to the absorption of the Micro-LED itself [37]. Considering the absorption losses of the Micro-LEDs, high reflectivity mirrors, such as metal mirrors or a large stack DBR, should be used on the non-emitting surface to prevent light leakage. On the emitting surface, mirrors with moderate reflectivity should be used to prevent the excessive back-and-forth reflection of light and minimize absorption losses.

Bottom reflector technology can reflect the Micro-LED active layer emitted light and the top fully reflected light back to the top, increasing the efficiency of the emitted light. Bai et al. proposed the integration of lattice-matched DBR at the bottom of the multi-quantum wells (MQWs), shown in Figure 6a [38]. The bottom DBR consists of 11 pairs of lattice-matched nanoporous (NP) GaN/undoped GaN. NP-GaN was obtained by electrochemical (EC) etching of the n^++^-GaN layer. The addition of the bottom mirror prevents light from leaking out on the bottom substrate, allowing more light to be reflected and emitted from the top to improve the LEE. Figure 6b presents a planar scanning electron microscopy (SEM) image of a Micro-LED wafer with a diameter of 3.6 µm and a spacing of 2 µm. The team developed a method to manufacture Micro-LEDs using selective overgrowth, which avoided the side wall damage caused by dry etching. A 9% ultra-high EQE was eventually obtained when the working wavelength was 500 nm, and the spectral width of the luminescence was reduced to 25 nm.

Regarding adding the bottom reflector LEDs, adding the top reflector can form the F-P cavity. The cavity makes the LED that is emitting light carry out constructive interference and destructive interference in the cavity, which makes the emitted light more collimated after the resonance selection. Huang et al. introduced the F-P cavity into GaN-based Micro-LEDs to improve collimation [18]. The RC Micro-LED structure is shown in Figure 6c. Both the bottom mirror and the top mirror of the F-P cavity are composed of SiO_2_/TiO_2_ DBR, but the logarithm is different. In this paper, the reflectivity changes in DBR with different logarithms at 450 nm were studied. Figure 6d shows that the more logarithms there are, the higher the reflectivity. The effect of bottom and top DBR reflectivity on the enhancement factor of light extraction was also studied. This provides an idea for how to set up a suitable cavity structure in Micro-LEDs. As shown in Figure 6e, when the working wavelength is 450 nm, the logarithm of DBR at the bottom is at least 10 pairs to make the reflectivity close to 100%, and the logarithm of DBR at the top is 3 pairs to make the reflectivity 65%. The F-P cavity constructed in this way can ensure light extraction when the wavelength satisfying the microcavity effect is resonated multiple times, and multiple resonant selections also make the FWHM of the emission spectrum narrow. GaN-based RC Micro-LEDs with a divergence angle of 78.7° and spectral width of 6.8 nm were made. More emitted light along the normal direction also gives the device great potential in AR applications, such as AR glasses and AR Head Up Display. Because of the resonance selection of the resonator, the output spectrum is narrowed, and the emission is directed.

In addition to applications in Micro-LEDs, the resonator can also enhance the LEE in OLEDs, PeLEDs, and quantum dot (QD) LEDs. In 2020, Joo et al. implemented an OLED display technology of over 10,000 pixels per inch by introducing an F-P cavity. The F-P cavity consists of a metasurface Ag mirror as a bottom mirror and an Ag electrode as a top mirror [39], shown in Figure 7a. The introduction of an F-P cavity reduces the divergence angle of OLEDs. The results in Figure 7b,c show that the luminous intensity and IQE of OLEDs with a metasurface F-P cavity are significantly higher than that of white OLEDs with a color filter. The three colors blue, green, and red represent blue light, green light, and red light, respectively. In addition to the color filter itself will reduce the OLED luminous intensity, the F-P cavity will carry out constructive interference on the light wave meeting the resonance condition of the microcavity to increase the intensity of the emitted light. In addition, by changing the diameter and density of nanocolumns in the bottom Ag mirror, different wavelengths of light can be enhanced by resonance to achieve color changes. In this way, OLED displays with smaller pixels can be achieved. Recently, Liang et al. introduced resonators into PeLEDs and developed a resonance enhancer cycling method [40], enabling the color of the emitted light to reach unprecedented purity after resonance selection, which provides a wide color gamut for the development of next-generation displays.

**Table 2 micromachines-14-01354-t002:** Summary of research on improving the emitted light collimation of LEDs.

Research Objective	LED Type	Wavelength	Optimized Structure Model	Simulation or Experiment	Ref.
Improve LEE and enhance spectral narrowing	Micro-LED	450~660 nm	Bottom NP GaN/undoped GaN DBR reflectors	Experiment	[38]
Improve LEE and enhance spectral narrowing	Micro-LED	440~460 nm	SiO_2_/TiO_2_ DBR reflectors F-P cavity	Simulation and experiment	[18]
Improve LEE and reduce pixel size	OLED	370~700 nm	Metasurface Ag and flat Ag mirrors F-P cavity	Simulation and experiment	[39]
Enhance spectral narrowing to improve color purity	PeLED	450~650 nm	Au mirror and DBR reflectors F-P cavity	Experiment	[40]

## 4. Control of the Deflection of Light Angle

The metasurface structure can not only improve the LEE of Micro-LEDs, but also accurately control the phase of the wavefront to realize the control of the emitted light angle of Micro-LEDs. The nanocolumn array is integrated on the top surface of Micro-LEDs, which is composed of nanocolumns with different diameters in one cycle to achieve 0–2π phase coverage. The phase of nanocolumns can be changed by changing the height of nanocolumns and the equivalent refractive index, which can be achieved by changing the diameter of nanocolumns. In the finite-difference time-domain method, the relationship between the diameter and phase of nanocolumns is obtained by scanning nanocolumns of fixed height and different diameters. The phase coverage of 2π is realized by selecting suitable nanocolumns with different diameters and adjusting different periods to achieve different emission angles. However, only coherent wavefronts can be controlled by metasurface structures, while Micro-LED emission meets the Lambertian-shaped emission and has poor coherence. Therefore, a resonator structure is added to Micro-LEDs to enhance the coherence of the emitted light and realize the metasurface structure to control the angle of the emitted light.

In 2018, Liu et al. demonstrated for the first time the control of metasurface structure on light emitted by LEDs with a wavelength of 460 nm through simulation [13]. This paper compared the far-field patterns of LEDs with directly added TiO_2_ nanocolumn arrays and resonant cavity LEDs (RCLEDs) with TiO_2_ nanocolumn arrays added to their surfaces. RCLEDs could achieve a 20° angular deflection by controlling the phase with metasurface structures, while LEDs could not. The result indicates that the RC structure can collimate the Lambertian-shaped emission of LEDs, approximating a plane wave, and enhance the spatial coherence of the emitted light from Micro-LEDs. Two years later, the team added the metasurface RCLED design to the GaP LEDs [41], proposing that as long as the reflectivity of the metasurface is low, resonators and metasurface can be designed separately and independently, showing the flexibility and universality of the metasurface design. In the GaP metasurface RCLEDs, the resonator consists of bottom Au mirrors and top DBR mirrors, with top nanocolumns selected with high-index Si materials. However, Si material has high absorption in the visible band, which will reduce the light output efficiency, so TiO_2_ material with a high refractive index can be chosen to replace it. In this paper, the 30° deflection of the emitted light beam with a wavelength of 620 nm was obtained (see Figure 8d). The far-field patterns of the experimental results correspond well with the simulation results. Figure 8 shows the full process of angular control through the integration of metasurface structures and RCLEDs.

In recent years, the demand for near-eye display devices has been increasing, and Micro-LEDs that can be emitted in unidirectional directions are undoubtedly one of the most ideal light sources, as unidirectional emissions can reduce optical crosstalk and provide a better visual experience. Huang et al. proposed a metasurface RC Micro-LEDs with unidirectional emission [19]. Different diameters of nanocolumns can produce different phase changes, which can satisfy the phase coverage of 0–2π within a period. The nanocolumns with different diameters are selected to form different periods to achieve unidirectional emission of light at different angles. The structure diagram and control phase schematic diagram are shown in Figure 9a,b, respectively. Controllable unidirectional emission Micro-LEDs can prevent light leakage. In a 3D display, unidirectional emission of light to the left eye and the right eye can reduce pixel crosstalk and improve the 3D display effect, shown in Figure 9c. In addition, Micro-LEDs composed of a pixel with a variety of emission directions can produce a wider viewing angle and more 3D viewing points (see Figure 9d), so as to have a better 3D display experience, providing more impetus for the future development of near-eye 3D Micro-LED displays. The team later made a 3D display using Micro-LEDs emitted unidirectionally and compared it with a traditional 3D display [42]. Through comparison, the results showed that unidirectional emission Micro-LED pixels can effectively reduce optical crosstalk and improve the display effect.

The control of the angle of emitted light by metasurface structure has also been developed in QD LEDs, which brings a new scheme for optimizing QD LEDs. Park et al. showed the control of metasurface structures on the angles of light emitted by the colloidal quantum dot (CQD) RCLEDs [43]. The CQD RCLED structure is shown in Figure 10a with five pairs of DBR mirrors on both the bottom and top to form a resonator, with the top integrating TiO_2_ nanocolumns to allow angular deflection. In Figure 10b, the relationship between the diameter and phase (red lines) of TiO_2_ nanocolumns with heights of 700 nm and 300 nm is shown, respectively. The results present that the phase coverage of 0–2π cannot be achieved by the short nanocolumns, but using patterned TiO_2_ nanocolumns with a high aspect ratio is difficult. Therefore, the team selected TiO_2_ nanocolumns with low aspect ratios to control the angle of light by adjusting the period and phase within the period, thus reducing the difficulty of the manufacturing process. The black line represents the relationship between diameter and transmittance, and it can be observed that the high TiO_2_ nanorods experience a sharp drop at a diameter of 250 nm. The final simulated and experimentally measured deflection angles in this paper are both 20°, with only a 4 nm deviation in peak wavelength. Therefore, the experimental results highly match the simulation. Huang et al. conducted a simulation study on the exit angle control of QD LEDs [44]. Comparing the emission light direction diagrams of the Gaussian beam and electric dipole light sources in the RCLEDs formed by the bottom flattened Ag mirror and top DBR mirror, they found that the RC of the flattened Ag mirror and DBR mirror failed to collimate the QD LED emission of light, resulting in unrealized angle control. The team proposed that the circular patterned bottom Ag mirror and the top DBR mirror formed an RC to realize the emitted light collimation, and the different deflection angles were controlled by selecting nanocolumns of different diameters to form different cycles, shown in Figure 10c. This method provides a scheme for the manipulation of a single QD, and presents the structural design differences between QD LEDs and Micro-LEDs in angular deflection.

In Table 3, the research on metasurface structures controlling the direction of light emitted by Micro-LEDs is summarized. Recent research findings are mostly simulation results, as the manufacturing process of multi-layer DBR mirrors and metasurface is relatively complex, and the design and fabrication of metasurface structures still face many challenges and high costs. Therefore, there might still be a long way before their application in displays.

## 5. Control of Near-Eye Polarization

Polarized LEDs play an important role in 3D displays, providing linearly polarized (LP) and circularly polarized (CP) light. Currently, the most common way for LEDs to produce LP light is to use a grating structure. By changing the period, duty cycle, and thickness of the grating, the transmission of transverse magnetic (TM) wave and the reflection of transverse electric (TE) wave are controlled to realize the emission of LP light. In 2010, Zhang et al. further optimized the grating structure and proposed that adding a dielectric transition layer with a lower refractive index than the GaN layer between the GaN layer and metal grating could improve the polarization characteristics of LEDs, that is, the transmission of TM wave and ER [45]. Ma et al. added Al grating on a sapphire substrate to realize LEDs with LP light emitted from the back, and the polarization degree could reach 0.96 [46]. Huang et al. directly integrated subwavelength metal gratings on the p-GaN surface and achieved a high ER of 14.17 dB by optimizing the grating period, thickness, and width [47]. Although LP light is realized through a grating structure, the energy of the TE wave is lost, and the luminous efficiency is reduced. Zhang et al. proposed the coupling effect between MQWs and metal gratings to generate surface plasma to improve the radiation recombination rate and polarization degree [48]. The team’s approach was to etch the p-GaN layer into a grating structure and coat it with a layer of Al grating to achieve linear polarization, as shown in Figure 11a. This method improves the luminous efficiency, but it does not fundamentally solve the TE wave loss.

To solve the TE wave energy loss, studies have focused on the application of metasurface structures in polarized LEDs. Wang et al. presented an LED with integrated metasurface structures and dielectric/metal (D/M) double-layer grating structures [16], shown in Figure 11b. The top gratings are used to transmit the TM wave and reflect the TE wave to generate LP light. The bottom metasurface nanocolumns act as half-wave plates to convert TE waves into TM waves, thus reducing the loss of TE waves. Both the luminous efficiency and ER are improved. This method provides a new idea for high-efficiency LP LEDs. Zhou et al. later applied the idea of the interaction of metasurface and gratings to OLEDs and proposed the integrated D/M nanograting structure at the top and holographic metasurface structures of the nanospeckle image at the bottom (see Figure 11c(iv)), realizing a white OLEDs with high-efficiency linear polarization with ultra-high polarization ratio [49]. In Figure 11c, the comparison of the four structure diagrams shows that the holographic metasurface not only converts TE waves into TM waves to improve emission efficiency, but also changes the emission direction of reflected light so that more light is emitted in the escape cone.

In 3D displays, LP light requires that the left and right eyes remain at the same level in order to reduce crosstalk between the left and right images, which makes for a very bad viewing experience. CP light overcomes this difficulty, reduces optical crosstalk, and improves the 3D display effect. Gao et al. designed CP Micro-LEDs using metasurface and grating structures [15], and the structure diagram is shown in Figure 12a(i). Regarding RC Micro-LEDs, the Al grating structure is integrated at the bottom to transmit TM waves, reflect TE waves, and treat TE waves as emission light, which also leads to loss of TM wave energy and a decrease in luminescence efficiency. The nanobricks integrated on the top are used as a quarter-wave plate to convert LP light to CP light, which provides a new idea for the design of CP Micro-LEDs. The TE reflectivity (RTE) and TM reflectivity (RTM) are changed by adjusting the period and duty cycle of the bottom grating to obtain high ER, as shown in Figure 12a(ii–iv). As a tool to flexibly modulate the polarization of light, the wave plate can change LP light into CP light, but the problems of high efficiency and tunable phase delay need to be solved. In OLEDs, Wu et al., using liquid crystal polymer as a substrate, proposed to prepare ultra-thin, flexible, foldable, and stretchable wave plates by a water-soluble transfer method [50]. The wave plate can achieve a delay effect at any wavelength of the visible band and has a high transmittance; most of the light transmittance is above 95%. Therefore, the use of this wave plate in flexible OLEDs will not lose the emitted light. In addition to using a wave plate, Jia et al. developed micro-cavity CP OLEDs through chiral light emitting body [51], shown in Figure 12b, which not only provided a new idea for CP OLEDs, but also overcame the problem of a chiral light emitting body reducing EL. Chiral emitter OLEDs have two Ag layers at the top and bottom as two metal mirrors and electrodes of OLEDs. A thin layer of two-dimensional organic single crystal is embedded between Ag layers. The advantage of an organic single crystal is that the anisotropy of its refractive index will lead to a birefringent microcavity, which is conducive to the occurrence of the Rashba–Dresselhaus (RD) effect and changes the linear polarization into circular polarization. Finally, the study of metasurface structures on polarization control of LEDs is summarized in Table 4.

## 6. Conclusions

In this paper, the applications of metasurface structures integrated with Micro-LEDs are reviewed from three aspects: to improve the LEE, to achieve directional emission, and to realize LP light and CP light. The reason for increasing the LEE is that the addition of metasurface structures will reduce the total reflection consumption of LEDs at the air-contact surface, so that more light is emitted in the escape cone. Control of Lambertian-shaped emission LED light by metasurface structures requires the use of RC to increase spatial coherence. Meanwhile, the realization of LP and CP light further accelerates the use of Micro-LEDs in 3D near-eye displays. Metasurface structures have also been widely used in OLEDs, QD LEDs, and PeLEDs, and optimized results have been obtained to improve light extraction efficiency and directional emission. In summary, the integration of metasurface structures provides more possibilities for the structural design and performance optimization of Micro-LEDs.

## Figures and Tables

**Figure 1 micromachines-14-01354-f001:**
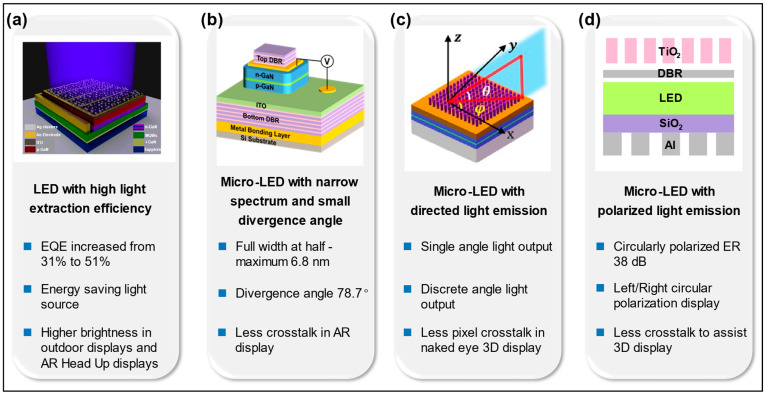
A list of four research directions for LEDs integrated with metasurface. (**a**) Disordered Ag nanoparticle metasurface to improve the LEE of LEDs. Reproduced with permission from [17], [*Light: Science & Applications*]; published by Nature Publishing Group, 2021. (**b**) Distributed Bragg reflector (DBR) resonator to improve the emitted light collimation of Micro-LEDs. Reproduced with permission from [18], [*Applied Physics Letters*]; published by AIP Publishing, 2022. (**c**) Metasurface structures to achieve directed light emission of Micro-LEDs. Adapted with permission from [19] © The Optical Society. (**d**) Metasurface and grating to improve the extinction ratio (ER) of polarized Micro-LEDs. Adapted with permission from [15] © The Optical Society.

**Figure 2 micromachines-14-01354-f002:**
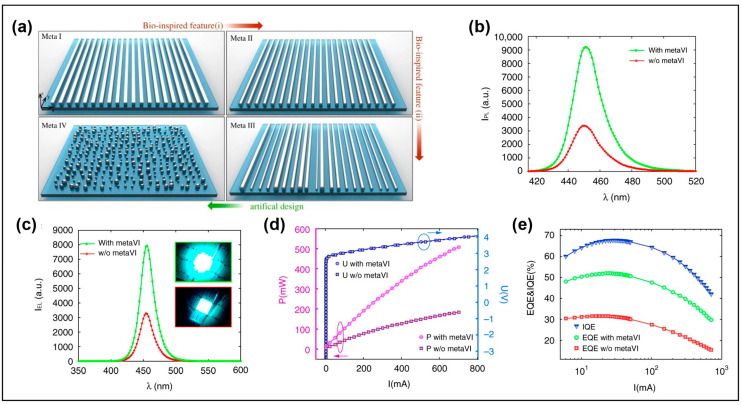
(**a**) Design process of disordered metasurface. (**b**) PL and (**c**) EL spectra of LEDs with and without Ag nanoparticles. (**d**) The output power and voltage of LEDs with and without Ag nanoparticles are a function of current. (**e**) IQE and EQE of LEDs with and without Ag nanoparticles. Reproduced with permission from [17], [*Light: Science & Applications*]; published by Nature Publishing Group, 2021.

**Figure 3 micromachines-14-01354-f003:**
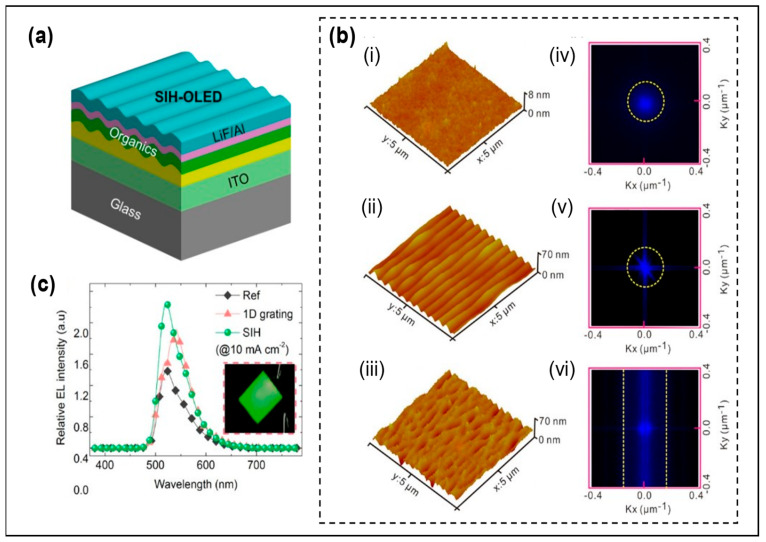
(**a**) Schematic diagram of SIH OLED device structure. (**b**) AFM images of (i) flat surface, (ii) one-dimensional grating patterned surface, and (iii) SIH patterned surface. (iv–vi) The corresponding fast Fourier transform patterns of the AFM images in (i–iii), respectively. (**c**) Relative EL spectrum perpendicular to the direction of the glass substrate. Reproduced with permission from [24], [*ACS Appl. Mater. Interfaces*]; published by American Chemical Society, 2016.

**Figure 4 micromachines-14-01354-f004:**
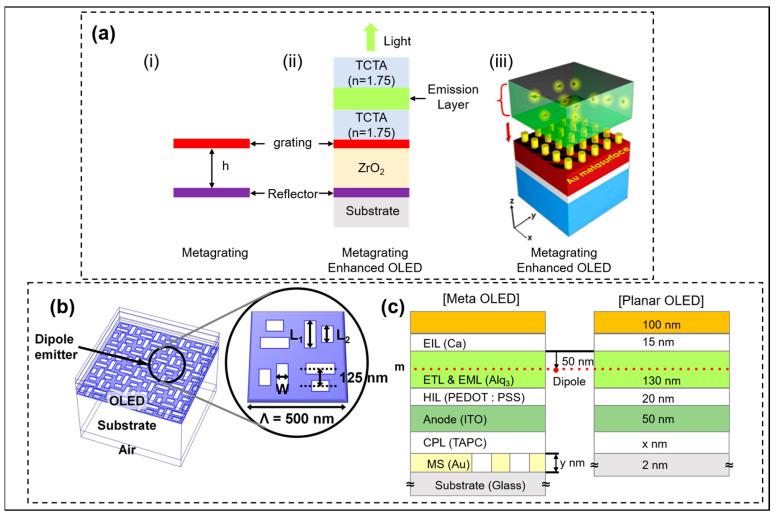
(**a**): (i) Schematic diagram of grating layer structure h height from the bottom mirror. (ii) Layer and (iii) 3D structure diagram of OLEDs integrated with metagrating. Reproduced with permission from [26], [*Applied Physics Letters*]; published by AIP Publishing, 2021. (**b**) 2D layer cross-section of nanoslot metasurface patterned and planar bottom emitting OLEDs. (**c**) 3D structure diagram of the bottom emitting OLEDs integrated nanoslot metasurface. The illustration shows one of the units of the nanoslot metasurface OLEDs. Reproduced with permission from [27], [*Scientific Reports*]; published by Nature Publishing Group, 2021.

**Figure 5 micromachines-14-01354-f005:**
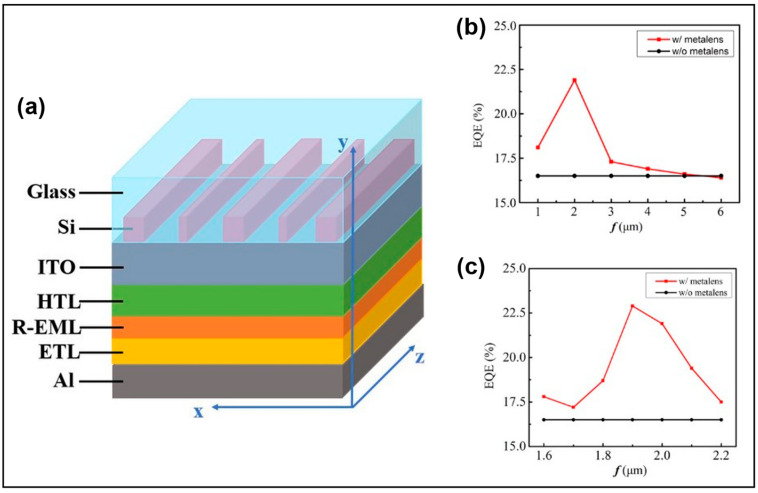
(**a**) A schematic diagram of the structure of Micro-OLEDs with metalens. (**b**) EQE with/without metalens at focal lengths (**b**) of 1 to 6 µm and (**c**) of 1.6 to 2.2 µm. Reproduced with permission from [28], [*Advanced Photonics Research*]; published by Wiley-VCH GmbH, 2021.

**Figure 6 micromachines-14-01354-f006:**
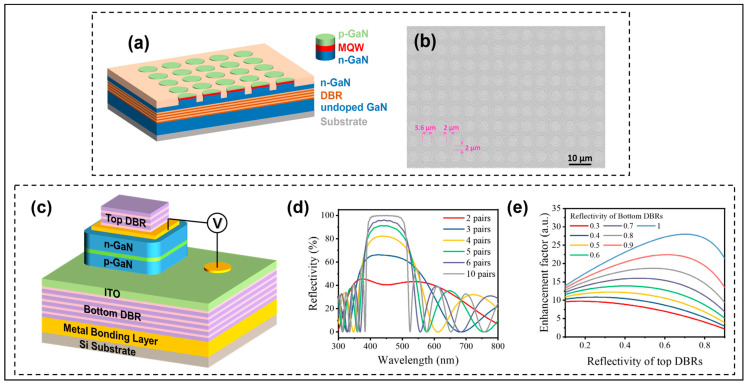
(**a**) Schematic diagram of the incorporation of Micro-LEDs and DBR. (**b**) Planar SEM image of a Micro-LED wafer. Reproduced with permission from [38], [*ACS Nano*]; published by American Chemical Society, 2020. (**c**) Schematic diagram of RC Micro-LED structure. (**d**) The effective reflectivity for different pairs of SiO_2_/TiO_2_ DBRs. (**e**) Relationship between top and bottom DBR reflectivity and light extraction enhancement factor. Reproduced with permission from [18], [*Applied Physics Letters*]; published by AIP Publishing, 2022.

**Figure 7 micromachines-14-01354-f007:**
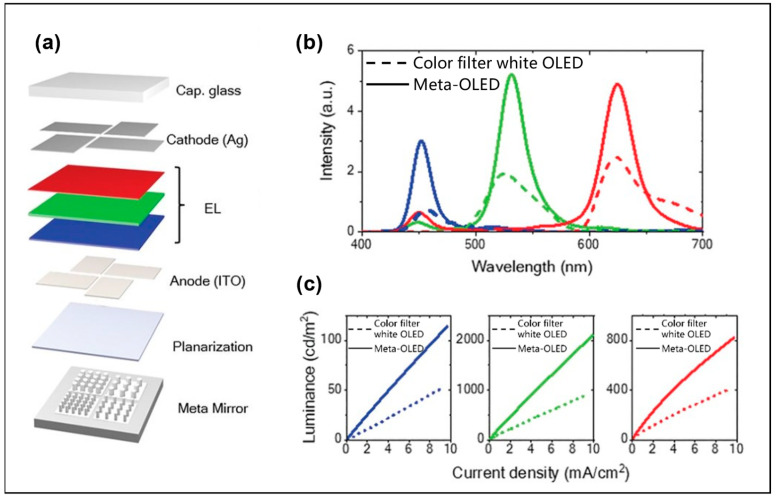
(**a**) Schematic diagram of meta-OLED design for metasurface mirror. (**b**) Electroluminescence (EL) spectrum and (**c**) luminance as a function of current density. Reproduced with permission from [39], [*Science*]; published by American Association for the Advancement of Science, 2020.

**Figure 8 micromachines-14-01354-f008:**
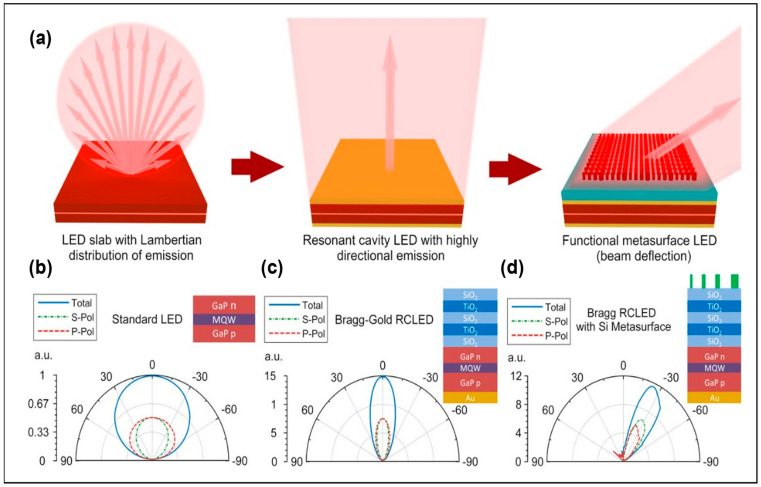
(**a**) Schematic diagram of metasurface control of LED-emitted light. The far-field diagram of (**b**) GaP LEDs, (**c**) Hybrid Bragg–gold RCLEDs, and (**d**) Hybrid RCLEDs with the integrated metasurface. Reproduced with permission from [41], [*Laser Photonics Reviews*]; published by Wiley-VCH GmbH, 2020.

**Figure 9 micromachines-14-01354-f009:**
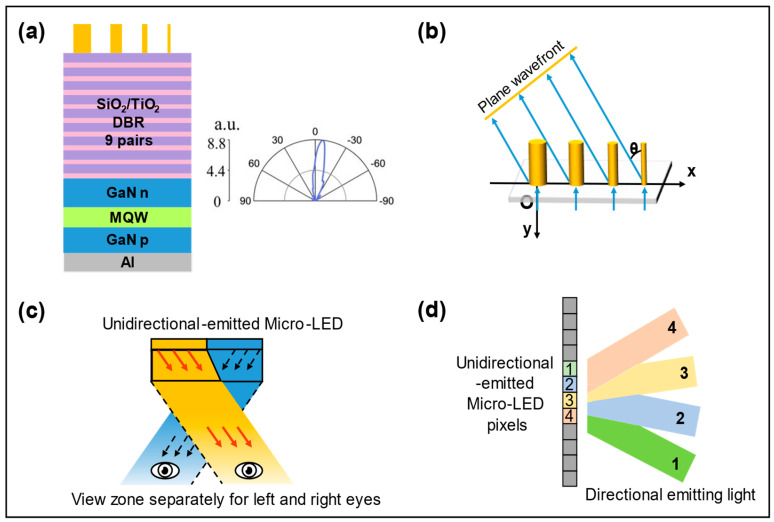
(**a**) Device structure and far-field diagram of Micro-LEDs with the beam deflecting metasurface. (**b**) Working schematic of the beam deflection supercell. (**c**) Unidirectional emission Micro-LEDs for naked-eye 3D display. (**d**) Multi-view naked-eye 3D display with unidirectional emission Micro-LED pixels. Reprinted/Adapted with permission from [19] © The Optical Society.

**Figure 10 micromachines-14-01354-f010:**
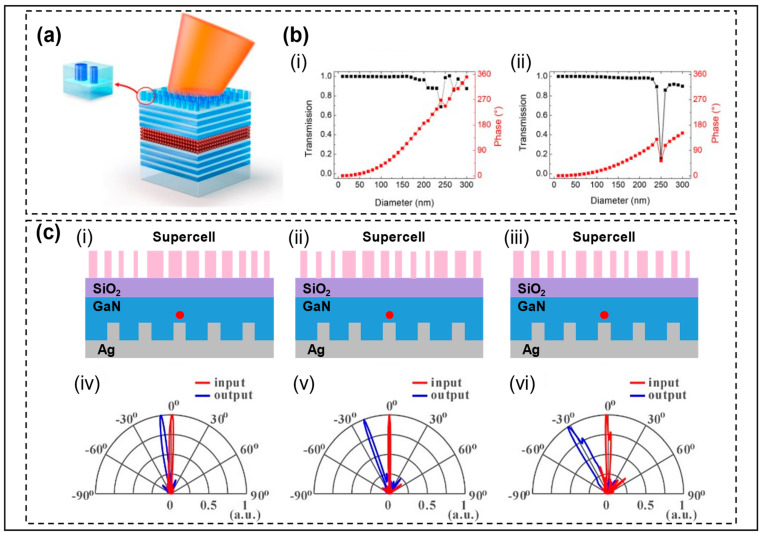
(**a**) Schematic of CQD RCLEDs with TiO_2_ metasurface. (**b**) Calculated look-up tables of a TiO_2_ nanocolumn with thicknesses of (i) 700 nm and (ii) 300 nm. Reproduced with permission from [43], [*Nanophotonics*]; published by De Gruyter, 2020. (**c**) (i–iii) Schematic structures of GaN-based QD LEDs with an Ag grating and a phase-gradient metasurface. Polar plots for the input beam and output beam of the metasurface. (iv–vi) are the cases for 10°, 20°, and 30° angle deflection, respectively. Reprinted/Adapted with permission from [44] © The Optical Society.

**Figure 11 micromachines-14-01354-f011:**
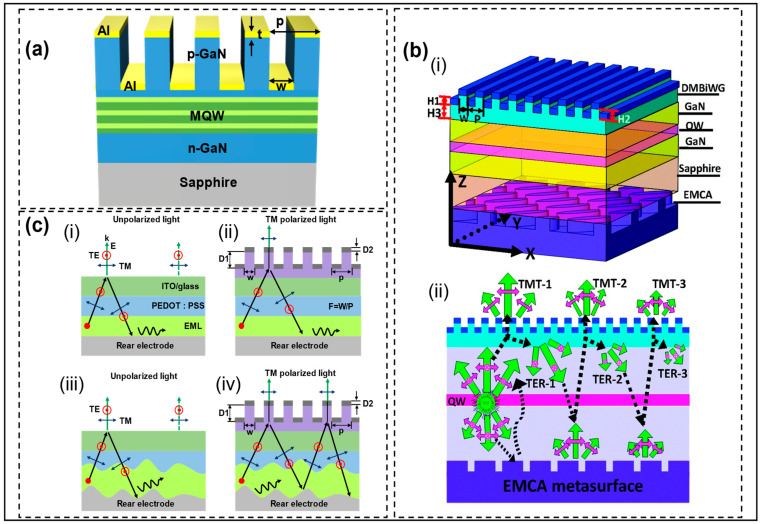
(**a**) Schematic diagram of the polarized LED structure. Reproduced with permission from [48], [*ACS Photonics*]; published by American Chemical Society, 2016. (**b**): (i) Structure diagram of InGaN/GaN LEDs with integrated metasurface and metal/dielectric grating structure. (ii) Propagation and polarization conversion processes of both TM and TE polarized components in the integrated structure. Reproduced with permission from [16], [*Nanoscale*]; published by Royal Society of Chemistry, 2017. (**c**) Schematic cross-section of device structures with (i) flat OLEDs, (ii) polarized OLEDs with D/M nanograting, (iii) OLEDs with holographic metasurface of nanospeckle image, and (iv) OLEDs integrated with D/M nanograting and holographic metasurface of nanospeckle image. Reproduced with permission from [49], [*Laser Photonics Reviews*]; published by Wiley-VCH GmbH, 2020.

**Figure 12 micromachines-14-01354-f012:**
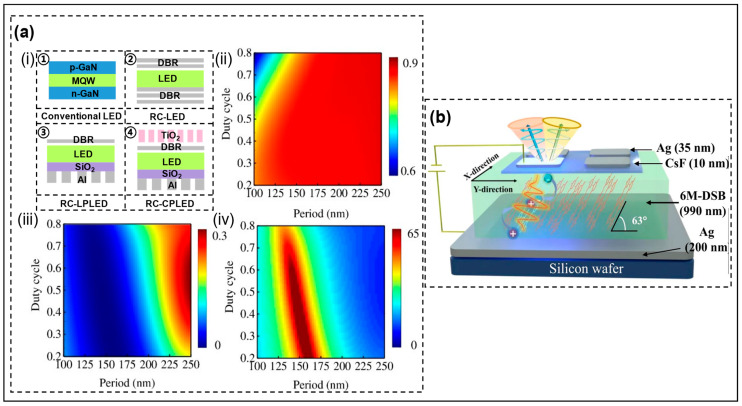
(**a**): (i) Cross-sectional views for different types of Micro-LEDs. (ii) RTE, (iii) RTM, and (iv) ER of Al nanograting with different periods and duty cycles calculated from numerical simulation. Reprinted/Adapted with permission from [15] © The Optical Society. (**b**) Schematic diagram of the microcavity CP OLED structure. Reproduced with permission from [51], [*Nature Communications*]; published by Nature Publishing Group, 2023.

**Table 1 micromachines-14-01354-t001:** Summary of research on metasurface structure to improve the LEE of LEDs.

Research Objective	LED Type	Wavelength	Optimized Structure Model	Simulation or Experiment	Ref.
Improve LEE	LED	440~470 nm	Disordered Ag nanoparticles	Simulation and experiment	[17]
Improve LEE	PeLED	780 nm	Nanobricks in the electron transport layer	Simulation	[23]
Improve LEE	OLED	440~570 nm	Speckle image holography metasurfaces	Simulation and experiment	[24]
Improve LEE	OLED	470~630 nm	Reflected supergrating	Simulation and experiment	[25,26]
Improve LEE	OLED	450~650 nm	Bottom nanoslot metasurface	Simulation and experiment	[27]
Improve LEE	Micro-OLED	640 nm	Metalens embedding the glass substrate	Simulation	[28]
Study the influence of wavelength and material on the SIH metasurface to improve LEE	OLED	440~670 nm	SIH metasurfaces	Simulation and experiment	[29]

**Table 3 micromachines-14-01354-t003:** Summary of research on metasurface structure to control LED emission angle.

Research Objective	LED Type	Wavelength	Optimized Structure Model	Simulation or Experiment	Ref.
Achieve light directional emission at the expected angle	LED	460 nm	Top TiO_2_ nanocolumns, DBR reflectors, and Al mirror cavity	Simulation	[13]
Achieve light directional emission at the expected angle	GaP LED	615~640 nm	Top Si nanocolumns, DBR reflectors, and Au mirror cavity	Simulation and experiment	[41]
Achieve light directional emission at the expected angle	CQD LED	580~620 nm	Top TiO_2_ nanocolumns and DBR reflectors cavity	Simulation and experiment	[43]
Achieve light directional emission at the expected angle	QD LED	520 nm	Top TiO_2_ nanocolumns and bottom circular patterned Ag grating	Simulation	[44]
Realize Micro-LED unidirectional emission	Micro-LED	445 nm	Top TiO_2_ nanocolumns, DBR reflectors, and Al mirror cavity	Simulation	[19]
Reduce pixel crosstalk by unidirectional emission Micro-LED	Micro-LED	460 nm	Top TiO_2_ nanocolumns, DBR reflectors, and Al mirror cavity	Simulation	[42]

**Table 4 micromachines-14-01354-t004:** Summary of research on metasurface structure to control LED polarization.

Research Objective	LED Type	Wavelength	ER or Polarization Degree	Optimized Structure Model	Simulation or Experiment	Ref.
Improve extinction ratio of LP light	LED	470 nm	60 dB	Dielectric transition grating	Simulation	[45]
Improve polarization degree of LP light	LED	445~470 nm	0.96	Wire-grid polarizer on sapphire	Simulation and experiment	[46]
Improve extinction ratio of LP light	Micro-LED	400~480 nm	14.17 dB	Subwavelength metal grating	Simulation and experiment	[47]
Improve polarization luminous efficiency and polarization degree of LP light	LED	500~540 nm	0.54	P-GaN and Al grating	Simulation and experiment	[48]
Improve polarization luminous efficiency and extinction ratio of LP light	LED	500~560 nm	20 dB	Top dielectric/metal grating and bottom elliptical metal nanocolumns	Simulation and experiment	[16]
Improve polarization luminous efficiency and extinction ratio of LP light	OLED	450~650 nm	17.8 dB	Top dielectric/metal grating and bottom holographic metasurface	Simulation and experiment	[49]
Obtain CP light	Micro-LED	450 nm	38 dB	Top TiO_2_ nanobricks and bottom Al grating	Simulation	[15]
Improve the luminous efficiency of CP light	OLED	450~700 nm	/	Waveplates based on de-ionized water immersion transfer method	Experiment	[50]
Improve the luminous efficiency of CP light	OLED	450~550 nm	/	Embedding a thin two-dimensional organic single crystal	Simulation and experiment	[51]

## Data Availability

Not applicable.

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
