# Peer review of "A Review on Micro-LED Display Integrating Metasurface Structures"

_micromachines, 2023, doi:10.3390/mi14071354_

Round 1

Reviewer 1 Report

This manuscript describes recent progresses on light emitting diodes integrated with metastructures. This manuscript is well-organized with proper reference papers, and its quality is publishable in this journal as it is.

One point I would like to note is that I recommend the authors to review the entire manuscript and correct the minor mistakes (typos) before publication.

None

Reviewer 2 Report

The author reviews the application of metasurface on micro-LED, OLED, and pLED. It can provide three benefits: increase the light extraction efficiency, control polarization of LED, and control the radiation pattern of LED. The review is comprehensive. However, the question is that the author reviews the metasurface on microLED, OLED and pLED together. However, the device structure of these device is very different. The thickness of OLED and pLED only a few hundred nanometers, but the LED is a few micrometers.  Therefore, the F-P effect (interference) is much more obvious in the OLED and pLED than the microLED devices. Moreover, the emission source of MQW in the microLED should be considered. In other words, the MQW in the micro-LED is not in the same height.  So, the interference is more complicated than the OLED or PLED design. For some micro-LED with strong cavity effect is mainly because they apply the thin film LED. It would be better if the author can point the difference between OLED and micro-LED out.

1. In Fig.2, the optimal design for the top and bottom DBR is related to the absorptoin of the LED itself. it is important to point it out.

2. In Fig. 3, please add the legend to the figure.

3. In Fig. 4, showing the experimental results or device performance is more attractive than showing cartoon of fabrication process.

4. In Table 1-3, for the experiment data, the wavelength should not be monowavelength. For example, Table1 (row 4), if there have experimental results, the wavelength should be provided. 

5. For Chapter 3, Could author comment why most of recent research is simulation results? What is the restriction to fulfill these designs? Could the simulation results highly match with the experimental results?

6. Extinction ratio should be a key factor to evaluate the performance of the polarized LED emission. 
For table 4 Please add the ER of each research if it is possible

Reviewer 3 Report

In this paper, authors reviewed the recent progress of Micro-LED display integrated with metasurface structures. Realization of polarized emission, high efficiency light extraction and controlled emission angle has been discussed. This review will be helpful for Micro-LED display society. I would like to suggest it published in Micromachines after being revised and polished. Here are some suggestions, hopefully authors can find it useful.

1.      It is suggested to link the corresponding display applications for these four research directions of Micro-LED integrated with metasurface in figure 1. And the reasons for such four directions should be discussed with respect to application conditions.

2.      In chapter 2.1, it might not be the simple reason for improving the LEE of LED to fabricate the F-P resonant cavity on Micro-LED. The emission direction can be modulated, and the FWHM of emission spectra can be narrowed. In some condition, amplified light emission can be achieved. Maybe it is not suitable to combine the F-P cavity in LEE chapter.

3.      Some statement should be revised. For example, it might not be fair to claim reduced manufacturing cost. It is still relatively hard to design and fabricate the metasurface. Based on current reports, there might be still a long way towards the applications in display.

4.      It might be strange to discuss matasurface on OLED in chapter 2.3. The topic is Micro-LED display. I believe there are many works need to be discussed in Micro-LED, at least Micro-OLED.

5.      More references are suggested.

Minor editing of English is suggested.

Round 2

Reviewer 2 Report

The revised manuscript address all the questions listed before.